



# Complex interplay of forcings drives Indian vegetation and summer monsoon variability during MIS 11

Dulce Oliviera[1,2], Stéphanie Desprat[3, 4], Qiuzhen Yin[5], Coralie Zorzi[1,2], Zhipeng Wu[5], Krishnamurthy Anupama[6], Srinivasan Prasad[6], Montserrat Alonso-García[7], Philippe Martinez[3]

[1] Centre of Marine Sciences (CCMAR/CIMAR LA), Universidade do Algarve, 8005-139 Faro, Portugal
[2] Divisão de Geologia e Georecursos Marinhos, Instituto Português do Mar e da Atmosfera (IPMA), 1495-165 Algés, Portugal
[3] Université de Bordeaux, CNRS, Bordeaux INP, UMR 5805 EPOC, F-33600 Pessac, France
[4] Ecole Pratique des Hautes Etudes (EPHE), PSL University, Paris, France
[5] Earth and Climate Research Center, Earth and Life Institute, Université Catholique de Louvain, Louvain-la-Neuve, Belgium
[6] Laboratory of Palynology and Paleoecology, Department of Ecology, French Institute of Pondicherry (IFP), UAR 3330, CNRS-MAEE, Pondicherry 605001, India
[7] Faculty of Sciences, Department of Geology (Paleontology), Universidad de Salamanca, Pza de los Caídos, 37008 Salamanca, Spain

*Correspondence to*: Dulce Oliveira (dulce.oliveira@ipma.pt , dulce_oliveiraa@hotmail.com)

**Abstract.** Marine Isotope Stage (MIS) 11 has long been considered a unique Quaternary interglacial due to its orbital similarities with the Holocene, persistence of high atmospheric $CO_2$ concentrations and extended duration triggering unusual polar ice-sheet loss. Despite its importance, Indian summer monsoon (ISM) variability within the core monsoon zone (CMZ), as well as its impacts on vulnerable tropical forests, remain unexplored. Here, we document, for the first time, MIS 11 ISM-driven vegetation changes and their underlying forcings by combining pollen analysis from IODP Site U1446, strategically retrieved from the Bay of Bengal to represent the CMZ, with model simulations. Our results reveal the distinct roles of insolation, $CO_2$, ice volume, and millennial-scale variability in driving coupled ISM-vegetation changes, depending on the changing boundary conditions through MIS 11. Orbital- and millennial-scale tropical forest changes mirror southern European vegetation and atmospheric methane variability, ultimately reflecting shifts in the Intertropical Convergence Zone (ITCZ) that impact the tropical regions, a primary source of $CH_4$ emissions.

Our proxy and model reconstructions show that ISM-vegetation changes during MIS 11c closely followed boreal summer insolation, revealing its dominant role under warm background conditions with high $CO_2$ and reduced ice volume. Conversely, during MIS 11b-a, ISM-vegetation decreased while insolation remained high, indicating that its influence was overshadowed by expanding ice sheets, lower $CO_2$, and the interaction of orbital and millennial-scale variations. Millennial-scale climate variability during the younger MIS 11b-a substages is expressed by prominent forest contractions tied to southward ITCZ shifts, Atlantic meridional overturning circulation (AMOC) reductions and high-latitude ice sheet dynamics, which were rapidly followed by abrupt forest expansions associated with northward ITCZ shifts, AMOC strengthening and $CH_4$ overshoots. Conspicuously, the first and most severe forest setback interrupted MIS 11 full interglacial conditions, suggesting that extreme ISM weakening could also occur under similarly warm future conditions. Our findings provide new insights into



ISM behavior during MIS 11, highlighting its high sensitivity to climate changes in the context of projected ISM intensification and its effect on the extent and composition of the tropical forest, which is key component of both global carbon and methane cycles.

## 1. Introduction

The Indian Summer Monsoon (ISM) stands as the dominant subsystem of the Asian summer monsoon (ASM) in terms of energy exchange, representing one of the strongest expressions of Earth's atmospheric-oceanic hydroclimate interactions. It accounts for up to 80-90% of Central India's annual rainfall in the so-called Core Monsoon Zone (CMZ), where it has its most representative expression and impacts millions of people and countless ecosystems (Gadgil, 2003). ISM-driven droughts, floods, landslides and other extreme events are among the most destructive natural hazards, causing widespread loss of life, severe socioeconomic disruption and irreversible ecosystems changes (e.g., IPCC 2022). With projections of an intensified global water cycle, including more frequent and severe extreme events, and persistent model uncertainty, predicting ISM variability remains a scientific priority (Turner and Slingo, 2009; Katzenberger et al., 2022). Inevitably, this leads to the study of ISM dynamics over geological times to assess its natural variability and the responses of vulnerable ecosystems, such as tropical forests, under globally warm interglacial conditions comparable to the present and near future.

Paleoclimate records of the two main Asian monsoon subsystems - the ISM and East Asian summer monsoon (EASM) – encompassing several or all key interglacials of the last 800 ka (Past Interglacials Working Group of PAGES, 2016) have relied on (1) Indian and Chinese speleothems, loess deposits and lake sediments (e.g., Zhisheng et al., 2011; Cheng et al., 2016; Kathayat et al., 2016; Sun et al., 2019; Zhang et al., 2019; Zhao et al., 2020; Gao et al., 2024), (2) marine sequences from the Arabian Sea, South China Sea and Bay of Bengal (e.g., Clemens and Prell, 2003; Ziegler et al., 2010; Caley et al., 2011; Bolton et al., 2013; Clemens et al., 2018, 2021; Gebregiorgis et al., 2018; Alonso-Garcia et al., 2019; Bhadra and Saraswat., 2022), and (3) model simulations, including data-model comparisons (Kutzbach, 1981; Kutzbach et al., 2008; Lyu et al., 2021; Sun et al., 2019; Sun et al., 2022). Since Kutzbach's (1981) seminal work on the Holocene, Northern Hemisphere summer insolation (NHSI) has been widely recognized as the dominant driver of orbital-scale ASM variability, principally due to its influence on land-ocean thermal gradients. Nonetheless, conflicting trends between these ASM records have generated extensive debate over the past three decades about the role of insolation versus internal forcings (see Cheng et al., 2022 for a recent review). Contrary to the "zero phase" view of a predominant precession rhythm and near-zero phase lag (e.g., Kutzbach et al., 2008; Cheng et al., 2016; Zhang et al., 2019; Bhadra and Saraswat., 2022), several studies have found dominant obliquity and eccentricity cycles, or combined 100, 41, and 23-kyr periodicities, and a precession phase lag of ~5-10 ka (e.g., Clemens and Prell, 2003; Ziegler et al., 2010; Caley et al., 2011; Bolton et al., 2013; Gebregiorgis et al., 2018; Clemens et al., 2021; Gao et al., 2024). These findings suggest a stronger influence of internal feedback mechanisms, related to global ice volume and atmospheric $CO_2$ concentrations, which may be linked to the transport of moisture from the Southern Hemisphere (e.g., Clemens et al., 2021). Beyond the complexity of deciphering these paradoxical results by evaluating which proxy and





model reconstructions most accurately capture ASM dynamics, recent studies from the India´s CMZ have reinforced the need

to differentiate the ISM and EASM subsystems mostly due to their distinct sensitivities to forcing mechanisms and moisture sources (Pan et al., 2017; Nilsson-Kerr et al., 2019; 2021). Yet, compared to the EASM, ISM monsoon rainfall is considerably less documented, especially within India´s CMZ where evidence from interglacials older than the Holocene is restricted to the long marine sequence of IODP Site U1446, located on northeast Indian margin. Records from this site, based on leaf wax isotopes, SST, planktonic and benthic seawater $\delta^{18}O$, X-ray fluorescence (McGrath et al., 2021; Clemens et al., 2021), and

planktic foraminifera assemblages (Bhadra and Saraswat, 2022), primarily focus on orbital-scale ISM variations across late Pleistocene glacial-interglacial cycles, which precludes a precise characterization of the ISM's response solely to interglacial boundary conditions. To date, past interglacial ISM variability in the CMZ has only been explored through comparisons between the Holocene and MIS 5e. Vegetation and salinity reconstructions from Site U1446 indicate stronger ISM activity during MIS 5e due to higher insolation (Clément et al., 2024). In contrast, regional data-model comparisons show that ASM,

including the core ISM region, was the only boreal monsoon system with a lower MIS 5e peak compared to the Holocene (Nilsson-Kerr et al., 2021), in line with multiproxy data from the northern Bay of Bengal (Wang et al., 2012). These inconsistencies highlight the complexity of proxy sensitivity and the heterogeneous responses of monsoonal subsystems to various forcings, underscoring the need for more targeted proxy reconstructions and data-model comparisons within India's CMZ during past warm intervals.

Here we focus on MIS 11 (425-374 ka), a period extensively studied mainly owing to the potential astronomical analogy of its interglacial substage MIS 11c with the Holocene (see Berger and Loutre, 2002; Droxler et al., 2003; Candy et al. 2014; Past Interglacials Working Group of PAGES, 2016; Tzedakis et al., 2022 for a review). Moreover, it presents a unique combination of relevant characteristics for ongoing and projected climate change, including prolonged $CO_2$-driven climate warming, higher-than-present sea levels linked to the loss of Greenland and West Antarctic ice sheets, remarkable Arctic

warmth, sustained North Atlantic warming, an AMOC comparable or stronger-than-present, and persistent abrupt climate change throughout MIS 11 (e.g., Lisiecki and Raymo, 2005; Jouzel et al., 2007; Melles et al., 2012; Candy et al., 2014; Grant et al., 2014; Dutton et al., 2015; Yin and Berger, 2015; Snyder, 2016; Hu et al., 2024). The majority of MIS 11 research has targeted the mid- to high-latitudes of the Northern Hemisphere, with a notable focus on the response of European vegetation (Candy et al., 2014; 2024). In contrast, studies in the India's CMZ encompassing MIS 11 are limited to the long record of Site

U1446 (Clemens et al., 2021; McGrath et al., 2021; Bhadra and Saraswat, 2022).  In addition, three high-resolution speleothem records from nearby caves in China, although not within the CMZ, specifically examine MIS 11 regional monsoonal variability (Cheng et al., 2016; Zhao et al., 2019; Wang et al., 2023). An in-depth analysis of MIS 11 reveals that all these records exhibit marked dissimilarities throughout the interval, and that some lack the required temporal resolution for a robust assessment (e.g., average of ~4 kyr, Bhadra and Saraswat, 2022), which hampers the identification of a clear signal of the ISM. This study

presents the first reconstruction of ISM-driven vegetation changes during MIS 11 over India's CMZ based on marine pollen analysis at the Integrated Ocean Drilling Program (IODP) Site U1446. This approach is well-established for tracking changes in the tropical forest composition, which primarily reflect variations in ISM rainfall that might be inferred to be driven by



fluctuations in the mean position of the Intertropical Convergence Zone (Clément et al., 2024). To provide unprecedented information on the mechanisms driving India's tropical vegetation and monsoon across the changing boundary conditions of

MIS 11—insolation, CO₂, and ice sheets—we also conduct a data-model comparison by performing a series of experiments using the LOVECLIM Earth System Model.

## 2. Environmental setting and pollen signal

Site U1446 (19°5'N, 85°4'E) was recovered during IODP Expedition 353 "Indian Monsoon Rainfall" in the NW Bay of Bengal, offshore the Mahanadi River basin, at 1440 meter below sea level under pelagic/hemipelagic sedimentation and

without turbidite disturbance (Fig. 1). Its chronology, developed by Clemens et al. (2021) by correlating the benthic oxygen isotope ($\delta18Ob$) record to the LR04 stack, extends back to ~1.45 Myr with an average sedimentation rate of ~14 cm.kyr⁻¹. The age-depth model between MIS 12 and MIS 10 includes four tie points (Clemens et al., 2021), and we added one control point to better constrain the end of MIS 11 $\delta18Ob$ plateau (Table S1). This site was strategically drilled to capture the signal from central India's CMZ, which is considered a prime region for paleomonsoon studies since it receives 80-90% of annual rainfall

during the summer (June to September, mean of ~250 mm.month⁻¹) and represents the monsoon patterns across the entire Indian Peninsula (Gadgil, 2003).

The ISM variability is primarily determined by the interhemispheric migrations of the Intertropical Convergence Zone (ITCZ; commonly defined as the latitude of highest precipitation) over the equatorial region in response to the seasonal insolation cycle (e.g., Webster et al. 1998; Gadgil, 2003; Goswami and Chakravorty, 2017; Zhang and Wang, 2008; Schneider

et al., 2014). The northward movement of the core of the monsoon rainfall during the summer occurs due to an ITCZ shift to the warmer Northern Hemisphere and the amplified land-sea thermal contrast which enhances moisture supply to the monsoon region. Besides the primary role of the ITCZ, the ISM dynamics are modulated by tropical atmospheric and oceanic phenomena such as the El Niño Southern Oscillation and the Indian Ocean Dipole, though their impact remains highly debated (e.g., Kumar et al., 1999; Hrudya et al., 2020; Krishnamurthy and Goswami, 2000; Goswami and An, 2023).

Vegetation distribution and composition on the Indian Peninsula are mainly influenced by the amount of annual rainfall and the duration of the dry season, both regulated by ISM changes, and to a lesser extent by the mean temperature of the coldest month (Legris, 1963; Champion and Seth, 1968; Gunnel, 1997). Notwithstanding the current prevalence of anthropogenic landscapes, the potential natural vegetation of the Indian subcontinent ranges from desert and tropical dry savannas in the interior and western areas to subtropical and tropical moist deciduous, semi-evergreen and evergreen forests

in the SW and NE regions. The most humid forest type, the wet evergreen forest, is mostly located in the Western Ghats of the SW peninsula (annual rainfall >2300 mm.yr⁻¹), while xeric vegetation dominates the Thar Desert in the NW region (annual rainfall <500 mm.yr⁻¹). The Mahanadi catchment area, more specifically, encompasses four principal types of potential vegetation (Fig. 1): (1) semi-evergreen forests, found in limited coastal regions and humid lowlands of the lower Mahanadi basin under maritime influence that receive the highest regional precipitation (annual rainfall >2000 mm.yr⁻¹); (2) tropical





moist deciduous forests, developing at higher altitudes in the basin, typically between 300 and 750 m (annual rainfall 1300 to 2000 mm yr⁻¹); (3) tropical dry deciduous forests (savanna), thriving in lowland areas, mainly in the central plains of the watersheds with reduced humidity (annual rainfall 900 to 1300 mm.yr⁻¹); and (4) mangroves, largely influenced by edaphic factors, spreading across coastal environments, including deltaic areas, estuaries, and lagoons. Site U1446 is ideally located for pollen-based vegetation reconstructions of the Mahanadi watershed due to its proximity to the river mouth and the narrow

continental shelf (∼25–60 km wide), which ensure the rapid delivery of continental material, such as pollen grains, to the deep sea. To complement, marine sediments from the eastern Indian margin, including from this site, have demonstrated that their pollen signatures present an integrated regional image of the vegetation in India's CMZ and are a robust tool for reconstructing ISM-driven vegetation changes (Zorzi et al., 2015; 2022; Clément et al., 2024).

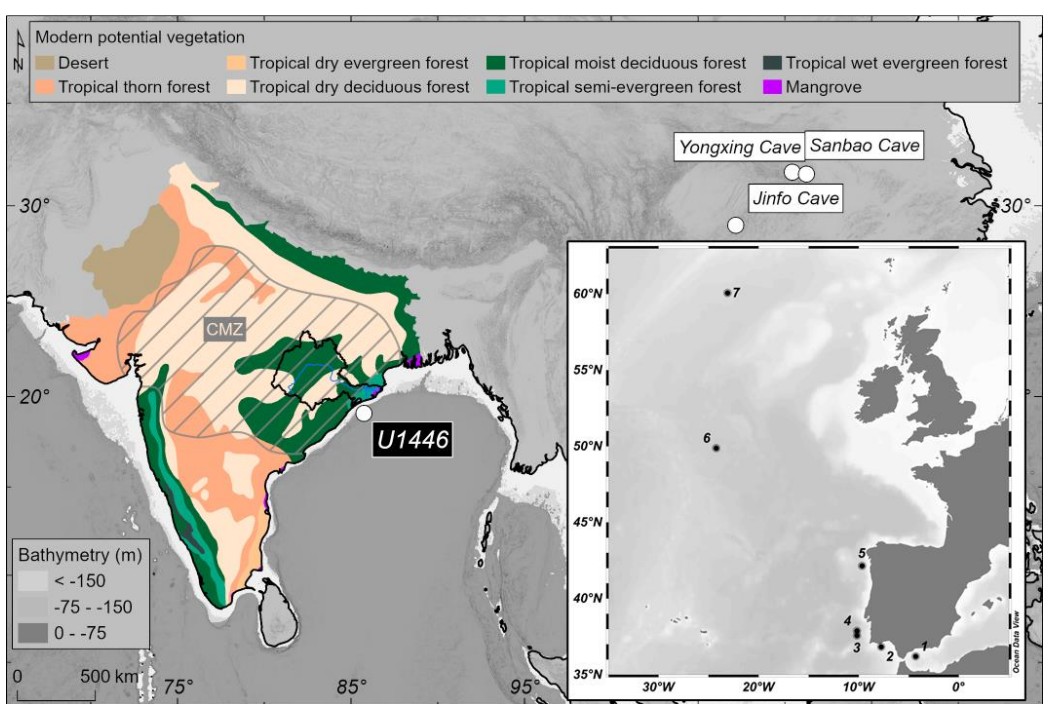

**Fig. 1.** Location of IODP Site U1446 and Chinese stalagmite records from Sanbao (Cheng et al., 2016), Jinfo (Wang et al., 2023), and Yongxing (Zhao et al., 2019) caves. Distribution of modern potential vegetation across the Indian Peninsula, with the Mahanadi catchment outlined in bold and the Core Monsoon Zone (CMZ) indicated by a striped area (adapted from Zorzi et al., 2022). Left inset: North Atlantic

records referenced in the text: 1. ODP 976 (Sassoon et al., 2023); 2. U1386 (Hes et al., 2022); 3. U1385 (Oliveira et al., 2016); 4. MD01-2447 (Tzedakis et al., 2009); 5. MD01-2443 (Desprat et al., 2005; 2017); 6. U1308 (Hodell et al., 2008); 7. ODP 983 (Barker et al., 2015; 2019).



## 3. Methods

A total of 62 levels were sampled for pollen analysis, 1.2- kyr mean temporal resolution, and prepared following the
155 methodology described in https://www.epoc.u-bordeaux.fr/index.php?lang=en&page=eq_paleo_pollens. Sediment subsamples were processed using the standard protocol for marine samples, which includes coarse sieving (150 μm mesh), consecutive treatments with cold HCl and cold HF, and micro-sieving (5 μm mesh). Pollen analyses were performed using a Nikon light microscope at ×500 and ×1000 (oil immersion) magnification, with identification based on tropical flora atlases (Thanikaimoni, 1987; Tissot et al., 1994; Gosling et al., 2013) and reference collections from the French Institute of
160 Pondicherry, India, and OSU OREME, France. Each sample comprised 18 to 53 pollen morphotypes to provide a reliable image of the vegetation community and its floristic diversity (McAndrew and King, 1976) and reached a total sporo-pollen sum between 142 and 247, with a main pollen sum >140 pollen grains excluding Himalayan taxa (*Alnus*, *Betula*, *Cedrus*, *Corylus, Myrica, Pinus*), aquatics and spores. Pollen percentages for terrestrial taxa were calculated from the main sum; Himalayan taxa from the main sum plus their individual counts; and aquatic and spores from the total sum (pollen + spores +
indeterminables + unknowns). Results were displayed in a pollen percentage diagram drawn with the software package Psimpoll 4.25 and Pscomb 1.03 (Bennett, 2008) (Fig. 2). Major pollen zones were identified by visual inspection of fluctuations in at least two ecologically distinct morphotypes (Birks and Birks, 1980), and confirmed through constrained hierarchical cluster analysis using the Euclidean distance between samples ("chclust" function from package *Rioja:* Juggins, 2009; R environment v. 3.1.1(R Core Team, 2014)) (Fig. 2).

Based on recent palynological analysis at Site U1446 (Clément et al., 2024), the interpretation of our results was supported by the main ecological groups of total tropical forest (TF), tropical deciduous forest, wet evergreen forest, grasslands and xerophytes (Figs. 2 and 3, Table 1)—each reflecting similar annual precipitation requirements (Legris, 1963; Champion and Seth, 1968; Bonnefille et al., 1999; Anupama et al., 2000; Barboni, 2000; Barboni and Bonnefille, 2001; Barboni et al., 2003). The expansion of the total tropical forest (TF), comprising all arboreal taxa except Himalayan morphotypes, indicates higher
forest cover primarily driven by intensified ISM rainfall. In contrast, the development of graminoid, mainly represented by Poaceae, reflects a more open environment due to reduced ISM, whereas the increase of xerophytes, mostly Amaranthaceae, Ephedraceae and *Artemisia*, suggests pronounced aridification. The tropical deciduous forest comprises species found in both dry and moist formations of the Mahanadi basin, mainly Combreataceae/Melatomataceae, *Haldina*, and *Glochidion* Shleichera, *Cassia*, *Holoptelea*, *Hardwickia*, and *Tectona*, which are grouped due to the indistinguishability of their pollen rain (Bonnefille
et al., 1999; Barboni et al., 2003). The wet evergreen forest includes pollen morphotypes from evergreen, semi-evergreen and moist deciduous forests which thrive in wettest forest formations in India. Its most representative pollen indicators are the taxa indicative of evergreen forests, namely *Elaeocarpus, Gnetum, Mallotus, Moraceae/Urticaceae, Olea paniculata, Syzygium*, and *Trema* (Bonnefille et al., 1999; Anupama et al., 2000; Barboni, 2000; Barboni and Bonnefille, 2001; Basumatary et al., 2014; Quamar and Bera, 2015; Mehrotra et al., 2022). Given the dependence of these taxa on high precipitation levels, this
group is used as the primary indicator of ISM rainfall variability.



Forest expansion and contraction events were statistically identified by the nonparametric regression technique LOWESS (locally weighted regression scatterplot smoothing) performed in the R environment ("Lowess" function from package *Stats:* Cleveland et al., 1992; R environment v. 3.1.1: R Core Team, 2014). The LOWESS function was employed to fit a smoothing curve ($\alpha = 0.2$) through the data and remove long-term variations by extracting the residuals (Fig. 4). Forest events were identified when at least one data point exceeded the threshold of one standard deviation ($1\sigma$) from the mean residual value of the wet evergreen forest percentages. Significant forest contractions occurring during the glacial inception were also associated with pronounced increases in xerophytic plant percentages that surpassed $1\sigma$ of their residuals.

To thoroughly evaluate the key drivers of MIS 11 vegetation and climate variability, we compared the pollen-based reconstructions with a set of MIS 11 transient simulations performed with the LOVECLIM1.3 model by Yin et al. (2021). These simulations are driven by time-varying insolation and $CO_2$ across two precessional cycles over MIS-11 and the model results, averaged over the India's CMZ (17°-27.5° N, 67.5°-90° E), are used for comparison (Fig. 5). LOVECLIM1.3 includes a dynamical vegetation module VECODE which simulates two plant functional types, tree and grass. Vegetation fractions within a land grid cell are dictated by empirical climatic conditions, specifically annual precipitation and positive degree-days (Brovkin et al., 1997). Previous studies using transient simulations with LOVECLIM1.3 have demonstrated consistency with vegetation reconstructions from Site U1446 for the Holocene and MIS 5e, and were further corroborated by the general circulation model HadCM3 (Clément et al., 2024).

## 4. Results and discussion

### 4.1. Indian vegetation and summer monsoon response to MIS 11 climate change

Cluster analysis of the pollen record distinguished nine pollen assemblage zones (PZs) that represent major shifts in the vegetation cover and composition between 438 and 363 ka, encompassing late MIS 12, the ~55-kyr long MIS 11 and early MIS 10 (Figs. 2 and Table 1 for a summary of the main features of the pollen zones). Based on modern relationships between the vegetation distribution and precipitation, shifts in the main ecological groups are interpreted as reflecting ISM rainfall changes, which in turn are thought to have been primarily controlled by the migration of the ITCZ's mean position and humidity transport (Clément et al., 2024, and references therein).









**Fig. 2.** Percentage pollen diagram of selected pollen morphotypes and ecological groups from Site U1446 in age (ka). Pollen percentage axis scale is consistent for all taxa. The dendrogram from the cluster analysis and the pollen zones (labeled as MIS 11 substage - pollen zone number within each substage) are displayed on the right. Green bands indicate major expansions of the total tropical forest, mainly driven
by the wet evergreen forest (zones MIS 11c-1 to 3, MIS 11a-1 and a-3), with the darkest green band representing their maximum expansion during MIS 11 (zone MIS 11c-2).

**Table 1.** Summary description of the pollen record of Site U1446 from late MIS 12 to early MIS 10 (this study). TF: Total tropical forest.

| Pollen zones (basal age in ka) | Pollen signature |
|---|---|
| **U1446-12-1** (437.2) | Dominance of xerophyte plants (Amaranthaceae, *Artemisia*, and Ephedraceae) with high values of herbaceous taxa, in particular of Cyperaceae and Poaceae. Very low frequencies of tree taxa (TF average < 4%) and mangrove. Maximum abundance of *Pinus*. |
| **U1446-11c-1** (427.4) | High-amplitude drop of xerophyte plants and herbaceous taxa associated with a pronounced increase in TF taxa percentages (TF average ~ 49.6%), mostly composed of *Mallotus*, *Olea paniculata*, *Cassia*, *Moraceae/Urticaceae*, *Glochidion* and *Haldina*, which is followed by a decreasing trend. Highest abundances of tropical deciduous taxa (*Cassia*, *Hardwickia*, *Ziziphus*). Rising percentages of mangrove taxa, constituted mainly by Rhizophoraceae. Decline in Himalayan taxa, primarily due to *Pinus*, which remains very low but fluctuate throughout the remaining pollen zones. |
| **U1446-11c-2** (418.6) | Marked decline in all xerophyte and herbaceous plants whereas TF taxa reach maximum percentages and pollen diversity (maximum of 58.2%) essentially due to wet evergreen taxa, specifically *Bombax*, *Celtis/Cannabis*, *Elaeocarpus*, *Gnetum*, *Hiptage*, *Macaranga*, *Mallotus*, *Moraceae/Urticaceae*, *Olea paniculata*, and *Trema*. Conversely, tropical deciduous taxa, including *Cassia*, *Hardwickia*, *Ziziphus*, fall to their minimums. Higher abundances of mangrove at the beginning, subsequently falling towards the top of the zone. |
| **U1446-11c-3** (408.3) | Increasing percentages of herbaceous taxa (mostly Cyperaceae and Poaceae) and tropical deciduous taxa (primarily caused by *Cassia*). Decreasing frequencies of TF taxa (average of 41.6%, mainly attributed to *Mallotus*) and Rhizophoraceae. |
| **U1446-11b-1** (396.1) | Significant drop in tropical forest taxa (TF minimum of 18.1%,), especially in all wet evergreen taxa, which contrasts with higher abundances of tropical deciduous forest taxa (*Cassia*, *Hardwickia*, *Ziziphus*). Increasing percentages of Poaceae coinciding with a rise in xerophyte plants (Amaranthaceae, *Artemisia*, and Ephedraceae). Lower abundances of mangrove taxa (Rhizophoraceae and *Sonneratia*). |
| **U1446-11a-1** (388.7) | Abrupt rise in TF taxa percentages (up to 42.3%), primarily owing to wet evergreen taxa (occurrences of *Bombax*, *Caryota*, *Elaeocarpus*, *Eurya*, *Gnetum*, *Hiptage*, *Moraceae/Urticaceae*, and *Trema* along with an increase in *Celtis/Cannabis*, *Mallotus*, and *Olea paniculata*) and Combretaceae/Melatomataceae. Decrease in tropical deciduous forest taxa, and in herbaceous and xerophyte plants (mostly Poaceae and Amaranthaceae, respectively). This pattern occurs twice due to a brief interruption in the middle of the zone forming an M-shape, where the trend reverses with a notable drop in TF taxa in favor of non-arboreal taxa. |



| | |
|---|---|
| **U1446-11a-2** (380.3) | Decrease of TF taxa percentages (mainly attributed to wet evergreen taxa; TF minimum of 9.1%) while those of herbaceous plants (Cyperaceae, Poaceae) and Amaranthaceae return to higher abundances. |
| **U1446-11a-3** (374.6) | TF taxa percentages increase (up to 26.7%), especially due to *Mallotus*, Combretaceae, and *Glochidium*, whilst herbaceous plants (Cyperaceae, Poaceae) and xerophyte plants decline. Slight increase in mangrove (Rhizophoraceae). |
| **U1446-10-1** (370.2) | Fall in all TF frequencies (TF minimum of 5.9%), including the disappearance of wet evergreen taxa, except of *Mallotus* and single occurrences of *Celtis/Cannabis* and *Olea paniculata*. In contrast, Poaceae and xerophyte plants (particularly Amaranthaceae and *Artemisia*) increase significantly. Low and relatively stable mangrove percentages. |

Glacial phases of MIS 12 and MIS 10 are characterized by open vegetation, mainly composed of grasslands and semi-arid plants (mostly Poaceae and Amaranthaceae) (PZs MIS 12-1 and MIS 10-1; Figs. 2 and 3, Table 1). The dominance of semi-arid steppe during these intervals suggests prevailing dry conditions and very low ISM rainfall. The MIS 11 in central India, from ~426 to 370 ka, comprises three major ISM-driven forest phases that are interspaced by intervals of more open vegetation (Figs. 2 and 3, Table 1). The longest, largest and most floristically diverse forest phase (TF: 32-58%) occurs during MIS 11c,

from ~427 to 396 ka (PZ MIS 11c- 1 to -3). This interval is essentially characterized by a well-developed wet evergreen forest (10-35%, mean 21%) indicating that a humid climate with increased summer monsoon rainfall persisted for approximately 31 ka. In addition, records from both TF and wet evergreen forest clearly display an asymmetric "M-shaped" pattern featuring a first peak in early MIS 11c and a stronger one in mid MIS 11c (~ 426 and 409 ka, respectively), followed by a gradual decline from late MIS 11c to MIS 11b (408.3-396.1 ka) (Fig. 3). ISM reaches its maximum intensity during mid-MIS 11c, coinciding

with the most significant expansion of TF and wet evergreen forest (wet evergreen forest peak: 35%, mean 26%), reduced abundance tropical deciduous forest and the weakest representation of both herbaceous plants, particularly Poaceae, and xerophyte plants (Fig. 3). The presence of highly moisture-demanding evergreen taxa, such as *Celtis/Cannabis, Caryota, Elaeocarpus*, *Eurya* and *Gnetum*, further supports our interpretation as they thrive in the heaviest rainfall regions of India (Fig. 2, Table 1). This wettest phase was rapidly interrupted by a major decline in the TF (from 58 to 32%) and followed by a gradual

contraction indicating a drying trend until the end of MIS 11c (PZ MIS 11c-3), although not below the climatic thresholds required to sustain the wet evergreen forest (mean 15%) (Figs. 2 and 3, Table 1). Over this interval, the tropical deciduous forest and grasses largely increased in the study area, suggesting an expansion of savannahs related to increasing dryness. Although the $\delta D_{precip}$ record from Site U1446 (McGrath et al., 2020; Clemens et al., 2021) lacks a corresponding decline during the first forest peak, the isotopic minima during the second peak indicate intensified monsoonal rainfall during mid-MIS 11c,

followed by a reduction through late MIS 11c, consistent with our vegetation reconstruction. During MIS 11b and 11a, from ~396 to 370 ka, Site U1446 shows a gradual TF decline with concurrent expansion of grasslands (Poaceae) and xeric elements



(mostly Amaranthaceae) reflecting a long-term drying in the CMZ (Figs. 2 and 3, Table 1). However, fluctuations in these ecological groups also reveal superimposed ISM variability. The cold substage MIS 11b (PZ MIS 11b-1, 396.1-388.7 ka) is marked by a strong increase in grasslands and xerophytes at the expense of the TF (minima of TF: 18% and wet evergreen

forest: 1.9%), indicating dry conditions due to reduced ISM. Subsequently, MIS 11a features two major expansions of the TF (PZ MIS 11a-1, 388.7-380.3 ka; -11a-3, 374.6-370.2 ka), which are separated by a more open vegetation phase that implies an ISM decrease (PZ MIS 11a-2, 380.3- 374.6 ka; maxima of TF: 42%, wet evergreen forest: 21%). The lower extent of the TF and wet evergreen forest during MIS 11a forest phases (maxima of TF: 9%, wet evergreen forest: 1.3%) compared to MIS 11c optimum (maxima of TF: 58%, wet evergreen forest: 35%), coupled with the increased abundance of herbaceous taxa, reveal

substantial less humid conditions (Fig. 3). The two forest expansion during MIS 11a coincide with decreases in δD values (Fig. 3). However, the $\delta D_{precip}$ does not record the distinct amplitudes of these peaks, likely due to its low time resolution as well as the influence of multiple factors affecting isotopic signatures, such as atmospheric transport pathways, source regions, and interactions with vegetation (McGrath et al., 2020; Clément et al., 2024).

Superimposed on the declining percentages of TF and wet evergreen forest, indicating gradual aridification through MIS

11, Site U1446 records suborbital-scale climate impacts on the Indian vegetation (Figs. 3 and 4). Based on the criteria outlined in Section 2 (Fig. 4), this short-term variability is captured by four major TF shifts (Fe-1 to Fe-4) that correspond to significant contraction/expansion events of the wet evergreen forest. These forest events reflect rapid transitions towards a drier/wetter climate, driven by ISM weakening/strengthening, respectively. The first and most dramatic forest contraction, recorded at ~408 ka, is marked by a high-amplitude TF decrease (decrease of 22.3% in ~1.5 kyr) mainly due to a decline of the wet

evergreen forest at the expense of grasses (Poaceae) (Figs. 3 and 4). After Fe-1, the progressive TF decrease during MIS 11b and -11a is interrupted by three significant forest events, Fe-2, -3, and -4, centered at ~388.7 ka, 383.3 ka and 374.6 ka, respectively (Figs. 3 and 4). These high-intensity events feature pronounced TF contractions rapidly followed by an abrupt expansion. Forest contractions are marked by large increases in xerophyte plants (maxima >21%) and concurrent declines in wet evergreen elements (minima <4%), reflecting an expansion of semi-arid steppe suggesting the most severe drought

conditions and reduced ISM during MIS 11. Conversely, subsequent high-amplitude forest expansions, recognized as sharp increases in tropical forest with wet evergreen elements (13% for Fe-2/Fe-3 and 23% for Fe-4, within 0.6-1 kyr), imply abrupt transitions to wetter conditions driven by stronger ISM.








**Fig. 3**. MIS 11 ISM-driven vegetation changes at Site U1446 - represented by the percentages of the main ecological groups/taxa (a) xerophyte plants, (b) Poaceae, (c) tropical deciduous forest, (d) wet evergreen forest, and (e) total tropical forest - in the context of changes in (f) U1446 $\delta D_{precip}$ (McGrath et al., 2020), (g) Chinese speleothem $\delta^{18}O$ from the Sanbao (Cheng et al., 2016) and Yongxing caves (Zhao et al., 2019), (h) Southern Iberian Mediterranean forest from Site U1386 and U1385 (bold line – 3 point moving average) (Oliveira et al., 2016; Hes et al., 2022) and (i) $CH_4$ concentrations from EDC Antarctic ice core (Nehrbass-Ahles et al., 2020). Pollen zones at the bottom distinguish three forest expansion phases (green bands) during MIS 11c and -11a substages, interspersed with phases of more open vegetation during MIS 11b and mid MIS 11a. Forest events (Fe-1 to -4) are depicted by brown bands indicating forest contractions/ISM weakening, while dashed lines represent forest expansions/ISM intensification. Arrows in (g) and (h) denote potential concurrent millennial-scale events.

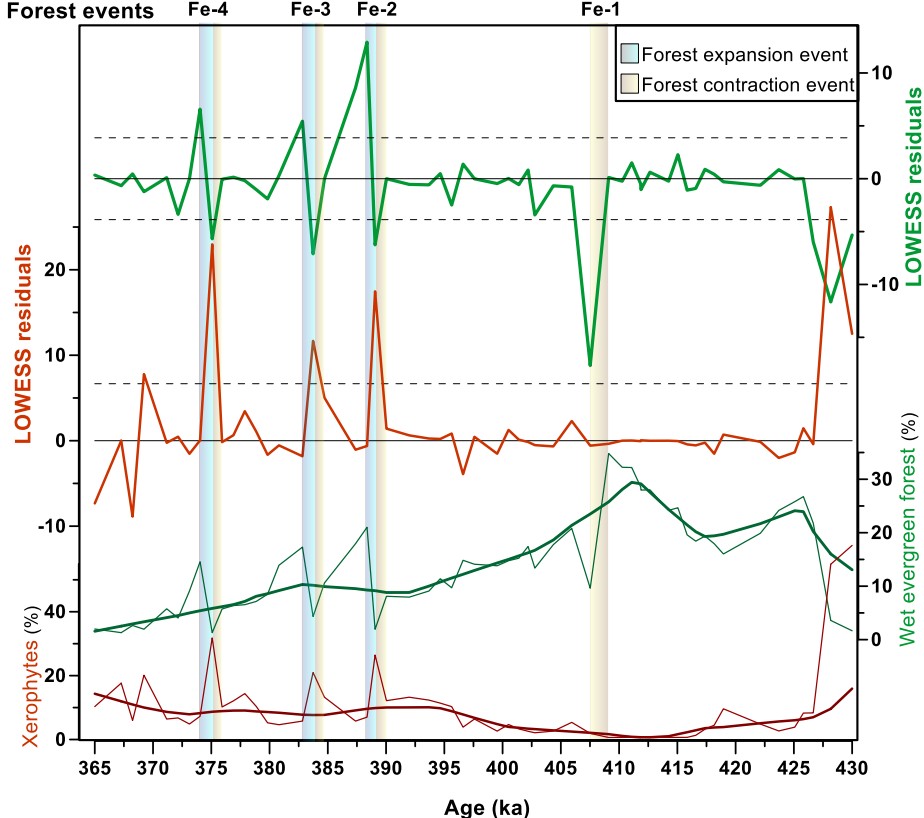

**Fig. 4.** Pollen percentage curves of the xerophyte plants and the wet evergreen forest (bottom panel) along with the respective LOWESS smoothing ($\alpha = 0.2$, bold line) and the residuals (top panel). Forest expansion and contraction events (green and brown bands, respectively) were identified when at least one sample of the model residuals exceeding the threshold of one standard deviation ($\pm 1\sigma$). Significant forest contractions occurring during the glacial inception were also concomitant with pronounced increases in xerophyte taxa percentages surpassing $1\sigma$ of their residual values.



## 4.2 Orbital-scale changes

### 4.2.1 Long-term changes in India's Core Monsoon Zone in a Northern Hemisphere context

Pollen-based reconstructions from Site U1446 reveal a sharp transition from a semi-arid steppe in late MIS 12 to established tropical forests in MIS 11 (TF: increase from 2 to 58%, Fig. 2 and 3), indicating a rapid response to an extreme hydroclimatic change. These results suggest that the vegetation in the CMZ responded to a steep northward shift of the ITCZ and associated intensification of the South Asian monsoon rainfall across the MIS 12/11 transition, in line with observations from the long records of the same site (Clemens et al., 2021), as well as from southern Bay of Bengal (BoB) (Bolton et al., 2013; Gebregiorgis et al., 2018) and East Asian speleothems (Cheng et al., 2016; Wang et al., 2023) (Fig. 3). A recent model-data comparison identified similar abrupt, high-amplitude forest expansions in mid-to-high latitudes of the Northern Hemisphere, whereas subtropical forests experienced a gradual and moderate growth, and tropical forests exhibited contrasting trends (Hes et al., 2022). Based on these results and the hypothesis that an unusually high global terrestrial biosphere delayed the $CO_2$ increase from MIS 12/11 transition until early MIS 11 (Brandon et al., 2020), Hes et al. (2022) proposed that temperate and boreal forests were the main contributors to the overall forest carbon sink during this interval. However, tropical forests exhibited varied and inconsistent trends, likely due to the scarcity of available records, along with low resolution and chronological uncertainties in the datasets used (Hes et al., 2022). Our pollen record reveals a previously undocumented contribution of tropical forests to the global terrestrial biosphere, highlighting the need to include more records from this region in future studies.

During MIS 11c, the pollen record from Site U1446 provides evidence for a large expansion of the tropical forest in India's CMZ, reflecting a sustained interval of increased ISM activity (Fig. 3). This forest expansion aligns with findings from previous palynological studies across the Northern Hemisphere documenting maximum vegetation cover and biodiversity during MIS 11c attributed to enhanced warmth and/or precipitation (e.g., Greenland: de Vernal & Hillaire-Marcel 2008; NE Russia: Melles et al., 2012; Siberia: Prokopenko et al., 2010; Central Asia: Xiao et al., 2010; Zhao et al., 2020; Hayashi et al., 2021; Europe: Candy et al., 2014; 2024 and references therein; North America: Fawcett et al., 2011; South America: Torres et al., 2013; NW Africa: Dupont and Agwu, 1992; Dupont et al., 1998; global data-model comparison: Kleinen et al., 2014). Our results further reveal that MIS 11c exhibits an asymmetric "M-shaped" structure, with a first forest peak in early MIS 11c and a more pronounced peak in mid-MIS 11c. This pattern implies two consecutive increases in ISM rainfall, with the second peak representing the largest expansion of wet evergreen forest, thereby indicating ISM reached maximum intensity between ~415 and 408 ka (Fig. 3). A tripartite division of MIS 11c with a mid-optimum is one of the key characteristics of this interglacial, being well-represented in records from Antarctic ice cores, global sea-level/ice volume, and both terrestrial and marine sequences across the Northern Hemisphere (e.g., Lisiecki and Raymo, 2005; Jouzel et al., 2007; Candy et al., 2014; 2024; Grant et al., 2014; Kandiano et al., 2017; Nehrbass-Ahles et al., 2020; Tzedakis et al., 2022 and references therein). However, the pattern and duration of each phase are heavily dependent on the archive and climate indicators under investigation. At close inspection, the asymmetric "M" pattern of our TF and wet evergreen forest records closely aligns, within age



uncertainties, with the high-resolution southern European marine pollen sequences from the Iberian margin (Desprat et al., 2005; 2017; Tzedakis et al. 2009; Oliveira et al., 2016; Hes et al., 2022), and the record of atmospheric methane concentrations

(CH$_4$ hereafter) from the EDC ice core (Nehrbass-Ahles et al., 2020) (Fig. 3). We propose that these concomitant changes in the vegetation and hydroclimate in the low and mid-latitudes reflect shifts in the ITCZ position that, in turn, have a prominent imprint on the CH$_4$ record. Such suggestion is supported by the ITCZ's well-recognized role in regulating past and present-day tropical monsoon systems and its impact on the extent of tropical wetlands, one of the largest natural sources of this greenhouse gas (e.g., Guo et al., 2012; Kirschke et al., 2013; Schneider et al., 2014; Bock et al., 2017). Secondary sources of CH$_4$ during

MIS 11c might have included boreal wetland expansion and/or permafrost thawing due to warmer conditions, as suggested by speleothem and paleotemperature data (Melles et al., 2012; Vaks et al., 2013, 2020; Batchelor et al., 2024). Unlike proxy records from low and mid-latitudes, the northern records show a plateau during MIS 11c or two comparable peaks, which resembles the slight difference between the two CH$_4$ maxima (Fig. 3). This similarity suggests that boreal regions may have also contributed to methane emissions during the early phase of MIS 11c, even though tropical wetlands remained the dominant

source throughout this substage. Tzedakis et al (2009) first attempted to link coupled southern European vegetation and CH$_4$ changes to shifts in the ITCZ, aiming to reveal the potential impact of low-latitude emissions on CH$_4$ over the past 800 ka. So far, confirming this hypothesis has been challenging due to the lack of detailed records of the low-latitude hydrological cycle for comparison with mid-latitude data. Subsequent studies (Nehrbass-Ahles et al., 2020; Tzedakis et al., 2022) have used the Sanbao speleothem record as a robust indicator of low-latitude hydroclimate and tropical wetland extent during MIS 11. Yet,

when compared with the CH$_4$ record, several distinct patterns emerge in terms of structure and phasing throughout MIS 11, particularly after 408 ka (Fig. 3). Additionally, the strong dissimilarities between the three speleothem records from the Chinese caves Sanbao, Jinfo and Yongxing (Cheng et al., 2016; Zhao et al., 2019; Wang et al., 2023) (Fig. 3) suggest that speleothem data alone cannot be used as a definitive proxy for low-latitude hydrological changes. Our ISM-driven vegetation reconstructions from India's CMZ provide, therefore, the missing evidence supporting Tzedakis et al (2009)'s hypothesis for

MIS 11.

During stadial MIS 11b, our reconstructions indicate an expansion of herbaceous savannah and semi-arid steppe reflecting reduced summer monsoon rainfall, followed by two major forest expansions during MIS 11a, interrupted by an open vegetation phase implying another ISM decline. Similar to Site U1446, pollen records across the Northern Hemisphere generally show the dominance of open vegetation during MIS 11b, indicating a shift to cooler/drier conditions, followed by a clear double-

350 peak pattern of forest expansion during MIS 11a, driven by a warmer/wetter climate. We infer that, akin to MIS 11c, the remarkable correspondence between low and mid-latitude vegetation and CH$_4$ changes during MIS 11b and 11a (Fig. 3) mirror variations in the hydrological cycle that are primarily controlled by the ITCZ's mean position. This climatic structure and patterns of MIS 11-b and 11-a are evident in sea-level/ice volume reconstructions and Antarctic ice core records, as well as in numerous palaeoclimate profiles of the Northern Hemisphere (e.g., Jouzel et al., 2007; Prokopenko et al., 2010; Fawcett et al.,

2011; Candy et al., 2014 and references therein; Grant et al., 2014; Barker et al., 2015; 2019; Nehrbass-Ahles et al., 2020; Sassoon et al., 2023), indicating a concomitant response to global climate change (Fig. 3). On a regional scale, increased aridity





during MIS 11b and mid MIS 11a intervals, are also recorded by the East Asian speleothems (Cheng et al., 2016; Zhao et al., 2019; Wang et al., 2023) (Fig. 3). Nevertheless, while most MIS 11 proxy-based precipitation reconstructions indicate higher humidity during the peak of MIS 11c compared to MIS 11a, a few monsoonal records, namely speleothem $\delta^{18}O$, show the

opposite trend (Fig. 3). These divergences likely arise from uncertainties regarding the proxies' exclusive signal of monsoon rainfall and their specific responses to forcing mechanisms or to heterogenous regional response of the Asian summer monsoon (e.g., Hu el al., 2019).

### 4.2.2 Interplay of forcings behind the ISM-induced vegetation changes

To understand the mechanisms driving long-term changes in India's CMZ across the changing boundary conditions of MIS 11, we compare our ISM-driven record of the wet evergreen forest with primary forcings - insolation, $CO_2$, and sea-level/ice volume – as well as with records from the mid- to high-latitude regions of the Northern Hemisphere (Figs. 3 and 5).





**Fig. 5.** MIS 11 drivers of ISM-driven vegetation variability on orbital and millennial timescales. North Atlantic changes: (a) meltwater discharges inferred from Si/Sr of Site U1308 (Hodell et al., 2008), b) ODP Site 983 polar planktic foraminifera (Barker et al., 2015; 2019),) and (c) $\delta^{13}C$ from Site U1385 indicative of AMOC changes (Nehrbass-Ahles et al., 2020). (d) $CH_4$ record (Nehrbass-Ahles et al., 2020) with red dots denoting predicted Dansgaard-Oeschger (D-O) warming events (Barker et al., 2011). U1446 data: (e) Percentage of the wet evergreen forest with forest events (Fe-1 to -4) (this study) and (f) LOVECLIM1.3 simulations for the tree fraction and annual precipitation (this study) with time-varying insolation and $CO_2$, and with fixed $CO_2$ at 280 ppmv (black line). Primary forcings: (g) July 21 insolation at 20°N (Berger and Loutre, 1991), (h) $CO_2$ concentrations from EDC Antarctic ice cores, arrows mark abrupt CDJ[+] (carbon dioxide jumps) coinciding with significant $CH_4$ increases (Nehrbass-Ahles et al., 2020) and (i) Relative sea-level from the Red Sea (Grant et al., 2014) and $\delta^{18}O_{benthic}$ from the LR04 (Lisiecki and Raymo, 2005).

The asymmetric M-shaped pattern during early to mid MIS 11c depicted by the wet evergreen forest curve of Site U1446 and the mid-latitude vegetation and $CH_4$ records (Fig. 3), provides evidence for the occurrence of shifts in the ITCZ's mean position that led to two humid phases separated by a multi-millennial drier interval. Both pattern and amplitude of the wet evergreen forest changes closely follow the precession-paced Northern Hemisphere summer insolation (NHSI) at 20°N (Fig. 5). Under warm interglacial conditions marked by high $CO_2$ and steady sea-level rise, the ISM intensifies during the two MIS 11c periods of high NHSI (precession minima at 409 and 428 ka) and weakens during the lower levels (precession maximum at 419 ka). Moreover, we reveal that the ISM and forest optima (wet evergreen forest max. of 35% at 409.1 ka) are nearly in-phase with the second and strongest NHSI peak of MIS 11 but precede the highest $CO_2$ levels and ice volume minimum (Fig. 5), which clearly emphasizes the dominant influence of insolation forcing in driving interglacial ISM–vegetation variability. The pivotal role of this forcing can be conceptually understood through its effect in driving the northward movement of the ITCZ and increasing moisture transport over the continent (e.g., Prell and Kutzbach, 1992; Kutzbach et al., 2008; Cheng et al., 2016; Schneider et al., 2014; Mohtadi et al., 2016; Gadgil, 2018; Jalihal et al., 2019). Maxima in summer insolation would have strengthened the thermal contrast between the Indian peninsula and the equatorial Indian Ocean, leading to intensified ISM rainfall via moisture transport from the Indian Ocean and Walker-type circulation anomalies. During late MIS 11c, the wet evergreen forest gradually declines in tandem with decreasing NHSI, while $CO_2$ levels remain high (> 270 ppm) (Fig. 5). This clearly indicates that insolation remained the dominant driver throughout the interglacial period, in agreement with findings from Clément et al. (2024), which demonstrate that $CO_2$ concentrations above ~250 ppm have a negligible impact on the vegetation dynamics in India's CMZ. Conversely, during the second part of MIS 11, the long-term wet evergreen forest changes diverge from NHSI trends and vary in concert with $CO_2$ and sea level/ice volume reconstructions (Fig. 5). These observations indicate that while the impact of the NHSI is dominant during MIS 11c interglacial conditions, $CO_2$ and ice-sheet dynamics become the primary drivers during the glacial inception, overriding the influence of insolation. The anti-phased patterns of the wet evergreen forest and NHSI changes during MIS 11b and mid-11a stadials suggest that the lower $CO_2$ concentrations (< 250 ppm) would have favored the replacement of tropical forests by herbaceous communities (Figs. 3 and 5). This suggestion is supported by studies on C3/C4 vegetation dynamics showing the photosynthetic advantages of C4 grasses over C3 vegetation in low $CO_2$ environments (Ehleringer et al., 1997; Cerling et al., 1998; Bond and Midgley, 2000). Moreover, larger northern hemisphere ice sheets would have decreased atmospheric humidity and its transport to the Indian peninsula, as





well as triggered an ITCZ southward shift (Lyu et al., 2021). As observed for the LGM, this shift would have led to increased aridity (e.g., Chiang and Bitz, 2005; Marzin et al., 2013; McGee et al., 2014) and a consequent reduction in forest cover in India's CMZ (Zorzi et al., 2022).

Taken together, our MIS 11 results from India's CMZ  do not support the "zero phase" hypothesis which suggests that the ISM variability is merely controlled by precession-paced NHSI (e.g., Prell and Kutzbach, 1992; Kathayat et al., 2016; Zhang et al., 2019); nor do they align with the view that it is solely driven by ice-sheet dynamics and/or $CO_2$ forcing (Caley et al., 2011; Bolton et al., 2013; Gebregiorgis et al., 2018; McGrath et al., 2021; Clemens et al., 2021; Yamamoto et al., 2022). Instead, our findings show that the controls of the coupled ISM-vegetation variability during MIS 11 are more complex and

their predominant role likely depends on the varying baseline climate conditions (insolation, $CO_2$ and ice-sheets). To ensure the robustness of these conclusions we performed LOVECLIM transient simulations forced by time-varying insolation and $CO_2$, as described in section 3. These experiments (Fig. 5) and linear regression analysis (Table S2) indicate that precipitation is the primary factor explaining the simulated tree fraction over India's CMZ, with surface air temperature playing a minor role, consistent with the model simulations by Sun et al. (2022). In agreement with the pollen data, both the simulated tree

fraction and precipitation closely track precession-paced NHSI during MIS 11c rather than the $CO_2$ record (Fig. 5). This divergence is particularly obvious during the $CO_2$ plateau in late MIS 11c, where both simulated variables decline gradually in response to NHSI (Fig. 5). These results are in line with previous studies demonstrating that the interglacial tree fraction over the northern tropical-subtropical regions including the Indian CMZ region, is primarily influenced by precession, while $CO_2$ has a minor effect (Yin and Berger, 2012; Su et al., 2022; Clément et al., 2024). Yet, in contrast to the pollen-based

reconstructions, the simulated tree fraction and precipitation continue to follow insolation trends across MIS 11b and early MIS 11a (Fig. 5). These discrepancies between pollen-based and model reconstructions may be largely attributed to the absence of ice-sheet forcing and associated freshwater flux in the simulations, as also proposed by data-model comparisons from the mid-latitudes of the Northern Hemisphere (Oliveira et al., 2018). The absence of dynamical ice sheets in the model would lead to an underrepresentation of millennial-scale variability and its interaction with orbital-scale changes, which might also account

for the data-model mismatches (see section 4.3.2). This millennial-orbital scale climate interactions would explain the observed abrupt forest decline during MIS 11c, Fe-1, and the subsequent forest events at the end of MIS 11b/transition to MIS 11a, Fe-2, which are inconsistent with the NHSI changes (Figs. 3 and 5).

    Noteworthy, according to our snapshot simulations, conditions in India's CMZ were considerably wetter and more forested at the climate optimum of MIS 11c, when NHSI occurs at perihelion (409 ka), than during the pre-industrial period (Fig. 3).

These results highlight the importance of further studying MIS 11 and the responses of terrestrial ecosystems in ISM-affected regions, as they offer valuable analogues for 21st-century climate conditions, where an intensified ISM is expected to lead to denser and greener vegetation, with deciduous trees potentially replacing grassland (Sharmila et al., 2015; Liu et al., 2020; Varghese et al., 2020; Katzenberger et al., 2021).





### 4.3 Suborbital-scale changes

#### 4.3.1. Impact of millennial-scale variability

The dramatic forest contraction Fe-1 occurs at ~408 ka during MIS 11c full interglacial conditions characterized by reduced ice volume and high $CO_2$ levels (Fig. 5). On a regional scale, an event analogous to this abrupt savannization of India's CMZ is only detected by the higher-resolution Chinese speleothem $\delta^{18}O$ record from Yongxing Cave, where it is identified as a weak monsoon event (Zhao et al., 2019) (Figs. 3 and 5). Within the chronological uncertainties, this event is well-documented in southern European marine pollen sequences, though it lacks a counterpart in the marine realm (Desprat et al., 2005, 2017; Tzedakis et al., 2009; Oliveira et al., 2016; Sassoon et al., 2023) (Figs. 3 and 5). As observed at both Site U1446 and Yongxing cave, it marks abruptly the end of MIS 11c optimum in southern Europe with no subsequent recovery to interglacial levels. Strikingly, unlike the sharp vegetation contractions, the $CH_4$ record only shows a slight decline (Fig. 3). These observations suggest that whilst tropical monsoon variability subtly influenced $CH_4$ emissions during this period, other sources may have played a larger role in the methane budget. Among the potential sources that warrant further study (e.g., Singarayer et al., 2011; Baumgartner et al., 2012; Guo et al., 2012), key contributors include high-latitude sources (wetland expansion, thawing permafrost, thermokarst lakes) driven by the warm conditions in boreal regions throughout MIS 11c, as well as increased emissions from tropical regions in the Southern Hemisphere.

The three high-amplitude and rapid TF expansions, Fe-2, -3, and -4, at ~388.7 ka, 383.3 ka and 374.6 ka, respectively, are marked by sharp rises of tropical forest dominated by wet evergreen elements (13% for Fe-2 and-3 and 23% for Fe-4, occurring in 0.6-1 kyr) (Figs. 3 and 5). These forest expansions imply abrupt shifts to wetter conditions driven by a stronger ISM and are concurrent with rising $CO_2$ concentrations and reduced ice sheet expansion. In addition, they all match with the predicted Greenland warming events derived from Antarctic temperatures (Barker et al., 2011) (Fig. 5). Despite being well imprinted in global records and multiple proxy reconstructions of the Northern Hemisphere, such as European vegetation records (Figs. 3 and 5), these millennial events have been scarcely explored, as most research has focused on the abrupt cool and dry events. On the other hand, significant forest contraction events coincide with lower $CO_2$ levels and increased ice volume during MIS 11b and -11a (Figs. 3 and 5). Numerous archives of the Northern Hemisphere provide a detailed characterization of their impact on both terrestrial and marine ecosystems, particularly the high-resolution reconstructions from the North Atlantic region, including the southern European pollen records (e.g., Oppo et al., 1998; Stein et al., 2009; Prokopenko et al., 2010; Rodrigues et al., 2011; Barker et al., 2015; Oliveira et al., 2016; Kousis et al., 2018; Hodell et al., 2023; Sassoon et al., 2023) (Figs. 3 and 5). Unlike these different lines of independent evidence, the East Asian speleothem records show contradictory evidence as the lowest resolution $\delta^{18}O$ profile of Sanbao cave (Cheng et al., 2016) only appears to record the youngest event, Fe-4, while the nearby higher resolution record from the Yongxing cave (Zhao et al., 2019), extending only until early MIS 11a, captures the events Fe-2 and Fe-3 (Fig. 3). On the other hand, the Jinfo cave (Wang et al., 2023), located further south, records all the events, despite considerable dating uncertainties. These differences in the speleothems $\delta^{18}O$ signatures may stem from differences in time resolution (Zhao et al., 2019), although a more convincing explanation involves distinct in-cave





processes (e.g., dripwater hydrology, carbonate dissolution, fractionation) and speleothem growth dynamics contributing to the final isotopic composition (Fairchild and Baker, 2012; Hu et al., 2019). Albeit our record's time resolution hinders assessing

the duration and abruptness of the ISM-driven forest events, it is evident that they remarkably correspond with drops to minimum $CH_4$ levels and subsequent high-amplitude overshoots (Fig. 3). These coupled changes further support the role of north-south ITCZ shifts in driving hydrological and vegetation changes in the low and mid-latitudes and their resulting imprint on the $CH_4$ record.

**4.3.2 Drivers of abrupt ISM changes and interaction with orbital-scale variability**

Conspicuously, the end of the MIS 11c optimum at ~408 ka is marked by an abrupt and high-amplitude forest contraction, Fe-1, that sharply contrasts with the gradual decrease in the NHSI, reduced ice volume and high and stable $CO_2$ concentrations (Figs. 3 and 5). We propose that this event is linked to its well-documented counterpart in soutprashern Europe (Fig. 3), which, due to the lack of an equivalent signal in the North Atlantic has been associated with increased aridity caused by a persistent

positive mode of the North Atlantic Oscillation (NAO) (Oliveira et al., 2016; Kousis et al., 2018). This correspondence between low and mid-latitude regions suggests a common response to atmospheric circulation changes, either independently or via teleconnections, that resulted in dry conditions in both regions. Given the current lack of a statistically significant link between the NAO and the ISM (Brönnimann, 2007), it is more likely that the co-occurrence of SW European and Fe-1 events is related to an atmospheric pattern resembling the El Niño–Southern Oscillation (ENSO). Recognized as one of the most important

global climate patterns, this coupled ocean-atmosphere mode has a well-known and strong impact on the ISM via changes in the Walker and regional monsoon Hadley circulation patterns (e.g., Webster et al., 1998; Krishnamurthy et al., 2000; Gadgil et al., 2007; Wang et al., 2013; Goswami and An, 2023). It exhibits two distinct phases, El Niño and La Niña, with El Niño phases typically associated with reduced ISM rainfall, while La Niña phases correspond to enhanced monsoon activity. Therefore, the Fe-1 event might have been driven by a shift to a prolonged or more frequent El Niño-like state. This shift

would have weakened the ISM, as proposed for weak monsoon events in Central India during the Holocene (Prasad et al., 2014; Riedel et al., 2021), ultimately causing a tropical forest decline. In line with our suggestion of a common link, several observational and model studies have demonstrated that ENSO also affects the circulation in the North Atlantic–European regions (Brönnimann, 2007). However, ENSO's impact on European climate and its mechanisms are still highly discussed, requiring extensive research to understand its potential role in triggering the forest events that interrupted MIS 11c full

interglacial conditions in both regions.

During the two MIS 11a interstadials, although the tropical forest maxima coincide with higher NHSI, $CO_2$ and sea-level, the changes are again too rapid to be solely attributed to these forcings (Fig. 5). The striking similarities and synchronicity, within age uncertainties, between the abrupt forest expansions Fe-2, -3, and -4, mid-latitude vegetation changes and sharp $CH_4$ rises suggest a rapid northward ITCZ shift (Fig. 3). Furthermore, these events correspond to Dansgaard-Oeschger (DO)



warming events, as predicted by the Greenland synthetic record of Barker et al. (2011), and abrupt carbon dioxide jumps (CDJ) identified by Nehrbass-Ahles et al. (2020) (Figs. 3 and 5). These results provide the first clear support for Nehrbass-Ahles et al (2020) recent hypothesis, which is based on the highest-resolution records of $CO_2$ and $CH_4$ from Antarctica EDC ice core. This study links CDJ and $CH_4$ overshoots to DO-like warming events driven by sudden AMOC intensifications, as inferred from centennial-scale benthic $\delta^{13}C$ records, such as those from Iberian Site U1385 (Fig. 5). AMOC strengthening would have

enhanced heat transport to the Northern Hemisphere, triggering a northward ITCZ shift and significant increase of low-latitude wetlands and $CH_4$ emissions, as corroborated by our findings. Noteworthy, Nehrbass-Ahles et al. (2020) used the record from Sanbao cave to indicate major ITCZ shifts; however, it only captures the most recent event, Fe-4, while our reconstructions provide new and consistent evidence of hydrologic change across all three events (Fig. 3). Forest contractions Fe-2, Fe-3, and Fe-4, driven by increased aridity in India's CMZ, correspond to three high-intensity, short-lived events that impacted both

terrestrial and marine ecosystems of the Northern Hemisphere. Frequently recognized as Heinrich-type events, these occurrences are associated with prominent cooling in the North Atlantic and peaks in Ice-Rafted Detritus (IRD) (e.g., Oppo et al., 1998; Stein et al., 2009; Prokopenko et al., 2010; Rodrigues et al., 2011; Candy et al., 2014; Barker et al., 2015; 2019; Oliveira et al., 2016; Kousis et al., 2018; Hodell et al., 2023; Sassoon et al., 2023). They are commonly linked to disruptions in the AMOC caused by meltwater discharges, as evidenced by concurrent reductions in Site U1385 $\delta^{13}C$ values and the

occurrence of IRD events (Fig. 5). Consistent with the mechanisms proposed for the MIS 11 forest expansions at Site U1446 and previous studies on the last glacial period in the Indian monsoon region (e.g., Mohtadi et al., 2014; Dutt et al., 2015; Ota et al., 2022; Zorzi et al., 2022), we infer that the forest setbacks are linked to AMOC disruptions that triggered a ITCZ southward shift and a consequent decrease in ISM-driven wet evergreen forest. Additionally, forest contractions at Site U1446 occur during periods of intermediate ice volume, once the $\delta^{18}O_b$ 3.5‰ threshold is exceeded, which aligns with studies

indicating that millennial-scale variability had a larger impact during medium ice sheet sizes, with iceberg pulses enhancing atmospheric-oceanic-land changes (McManus et al., 1999; Barker et al., 2015). These results, coupled with similar vegetation responses observed in the mid-latitudes (Fig. 3), highlight the important role of ice sheet dynamics in influencing the hydrological cycle and terrestrial ecosystems across different regions of the Northern Hemisphere.

The abrupt nature of these ISM-driven vegetation changes throughout MIS 11 emphasizes the need of considering the

interaction between long-term and millennial-scale changes, which has largely remained underexplored, to fully understand interglacial ISM-vegetation variability.

## 5. Conclusions

India's tropical vegetation and summer monsoon changes are investigated for the first time during MIS 11, a potential analogue of the Holocene, through pollen-based vegetation reconstructions from IODP Site U1446 and their comparison with

model simulations. The strong correspondence between our tropical forest record, southern European vegetation, and $CH_4$ concentrations suggests that the link between low and mid-latitudes was mediated by shifts in the ITCZ's mean position and



reflected in the $CH_4$ levels, largely controlled by low-latitude wetland emissions. We reveal that the dominant forcings of coupled ISM-vegetation variability, primarily driven by ITCZ shifts, depended on the interplay between the boundary conditions through MIS 11 - insolation, $CO_2$, and ice sheets - as well as orbital and millennial-scale changes related to the AMOC.

During interglacial MIS 11c, the asymmetric M-shaped pattern in our tropical forest record mirrors changes in $CH_4$, indicating shifts in the ITCZ closely tracking precession-paced NHSI. Unprecedented evidence shows that ISM and forest maxima occur in phase with the strongest NHSI peak of MIS 11c but precede the maxima in $CO_2$ and sea level, demonstrating the dominant influence of insolation forcing on ISM-vegetation variability under warm and high atmospheric $CO_2$ background conditions. This forest optimum is interrupted by the most prominent forest contraction, ~408 ka, evidencing that extreme reductions in ISM can also occur during globally warm conditions. An equal impact is observed in terrestrial ecosystems at mid-latitudes, despite no equivalent change in the marine realm or IN the $CH_4$ record. While the drivers behind this event remain unclear, we propose that future research should prioritize atmospheric circulation patterns affecting both low and mid-latitude regions, with a specific focus on ENSO.

The second part of MIS 11 is marked by ISM-vegetation changes that diverge from NHSI trends and align with $CO_2$ and sea level/ice volume reconstructions, revealing that atmospheric $CO_2$ and high-latitude ice-sheet dynamics become the primary drivers during substages MIS 11b and 11a. Moreover, superimposed on the long-term trends, our record captures three major ISM-driven forest events, at ~388.7 ka, 383.3 ka, and 374.6 ka, indicating that interactions between long-term and millennial-scale changes are a prominent feature during the glacial inception of MIS 11. Abrupt forest expansions suggest a rapid ITCZ poleward shift associated with $CH_4$ overshoots and correlated with invigorated AMOC activity. In contrast, forest contractions indicate a southward ITCZ shift linked to high-latitude processes involving ice rafting, North Atlantic cooling, and AMOC reductions.

Our transient experiments with time-varying insolation and $CO_2$ reveal that the NHSI is the primary driver of climate dynamics during MIS 11c, corroborating the findings from Site U1446 pollen-based reconstructions. The mismatch observed between the data and model outputs during MIS 11b and early MIS 11a emphasizes the need to incorporate ice sheet dynamics into model simulations in order to better simulate the interplay between orbital and millennial-scale variability.

Our pollen data and simulations indicate that the India's CMZ was substantially wetter and more forested during MIS 11 compared to the pre-industrial time, reinforcing therefore the importance of continued research on this past interglacial period and its ecological implications. In the context of future climate projection, it is crucial to account for the impact of an intensified ISM on vulnerable tropical forest dynamics and biodiversity.



**Data availability**

The data presented in this manuscript will be archived at Pangaea upon publication of the preprint and assignment of a DOI. Currently, the datasets are available as supplementary material for the review process.

**Author contributions**

D.O. conducted the pollen analyses, with contributions from S.D., S.P., K.A. and C.Z., and designed the research. Q.Y. and Z.W. performed the climate-model experiments. D.O. led the manuscript writing, with input from all authors. Funding acquisition by P.M., S.D. and D.O.

**Competing interests**

At least one of the (co-)authors is a member of the editorial board of Climate of the Past. The authors have no other competing interests to declare.

**Acknowledgements**

This research was funded by the Portuguese Foundation for Science and Technology (FCT) through projects INDRA (EXPL/CTA-CLI/0612/2021) and Hydroshifts (PTDC/CTA-CLI/4297/2021), CCMAR projects UIDB/04326/2020 (https://doi.org/10.54499/UIDB/04326/2020), UIDP/04326/2020 (https://doi.org/10.54499/UIDP/04326/2020), and CIMAR laboratory funding LA/P/0101/2020 (https://doi.org/10.54499/LA/P/0101/2020), and contracts for D. Oliveira (CEECIND/02208/2017) and C. Zorzi (CEECIND/CP1729/CT0001). Additional funding was provided from the French LEFE program (INSU-CNRS, project MICMAc) and IODP France. Computational resources were provided by Université Catholique de Louvain (CISM/UCL) and CÉCI, funded by F.R.S.-FNRS (convention 2.5020.11) and the Walloon Region. We thank the IODP Expedition 353 team, L. Devaux, M. Georget for technical support, and V. Hanquiez for Fig. 1.

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
