# Peer review of "Complex interplay of forcings drives Indian vegetation and summer monsoon variability during MIS 11"

_EGUsphere, 2024_

## Author Comment (AC1)

**Authors' Response to Reviewer #1:**

This article is a high-quality piece of work with some important new data spanning an interesting climatic transition with material being delivered from central India and providing critical new constraints on the development of environment during interglacial periods within the core monsoon area. I believe that it is worthy of publication but it will need some rewriting before it could be accepted. I actually found the paper really difficult to follow and understand. The writing is very dense and there is so much use of abbreviations that it makes it very difficult to follow. Not everyone is going to be as familiar with all these abbreviations as the authors and I found their use obstructive to my understanding of the manuscript. I'm sure the data is a good quality and if I understood this correctly it suggests that during warm periods there was a greater prevalence of tropical forest in central India compared to other times but that the intensity of the Indian monsoon and the vegetation is regulated by orbital processes but also the extent of ice sheets in high latitudes. One of the other things that stopped me understanding the manuscript more fully was the use of the MIS stages rather than telling me whether it's warmer or a colder period. Again, the number of the stages and sub stages are probably very familiar to the author but not to all the readers and I think it might be helpful if they were to try and better describe what each provide more information about what each stage number means. The authors assume too much prior knowledge. I was a little confused about the comparison of the climate model and the data itself. The conclusion appeared to be that the model wasn't very good unless you accounted also for glacial variations which at least in my reading suggests at the model isn't that good and so I wonder why so much time was spent discussing the results. I provide here below a number of smaller editorial comments and questions to help improve the understanding of the manuscript.

We thank the reviewer for the positive feedback and for the constructive comments. We will simplify the language and break down long, complex sentences to improve readability. We will also reduce the number of abbreviations throughout the manuscript, removing or spelling out non-standard or unnecessary ones. Additionally, we will provide clearer explanations for the Marine Isotope Stages (MIS), specifying their ages and indicating whether they correspond to warm or cold periods.

Furthermore, we acknowledge the reviewer's concern regarding the discussion of the climate model results. While we agree that the simulations are not fully comprehensive, they still provide valuable insight into the impact of insolation and $CO_2$ during the MIS-11 optimum period. Addressing all the hypotheses raised in this study would require more advanced simulations incorporating interactive ice sheets and related freshwater fluxes, which we recognize as direction for future research in the manuscript. We also recognize that the model results play a limited role in our analysis and do not require extensive discussion. Therefore, in the revised manuscript, we will reduce the text related to model interpretations while maintaining their key contributions.

Below, we address each of the reviewer's specific comments.

- Line 18 - Marine Isotope Stage (MIS) 11 - Should say how old this is in ka. We will add this information in the abstract.

- Line 20 - core monsoon zone (CMZ) -What is that. The core monsoon zone (CMZ) is the region most reflective of Indian summer monsoon rainfall variations as its variability closely mirrors the overall monsoon variability across India. We will improve its description in the introduction to enhance clarity.

- Line 28 - MIS 11c - Again, we need the ages of these substages. We will add this information in the revised manuscript.

- Line 36 - ISM weakening could also occur under similarly warm future conditions – Triggered by what? Be specific. Here we are referring to the monsoon changes triggered by global warming. We will revise this statement to improve clarity.

- Line 43 - Core Monsoon Zone (CMZ) – Label this on a map. The CMZ is already highlighted in Figure 1 with a striped area.

- Line 140 - rapid delivery of continental material – Depends what you mean by "rapid". Could be a whole 100 ky glacial cycle for coarse grained sediment. There is likely buffering in the flood plain for fine material including the pollen. In light of this comment, we will replace 'rapid delivery of continental material' with a more precise description. Pollen at Site U1446 is primarily transported by rivers, with the Mahanadi River as the dominant contributor (Zorzi et al., 2022; Clément et al., 2024). Over 90% of its annual sediment discharge occurs during the summer monsoon, efficiently carrying pollen offshore. Due to its hydrodynamic behavior, similar to fine silts and clays, pollen is transported with suspended sediments and exported beyond the shelf (e.g., Hooghiemstra et al., 2006). This process is favored by the region's narrow continental margin and submarine canyons, allowing pollen to rapidly sink through the water column (Raman and Reddy, 2001). These processes limit long-term buffering in floodplains, ensuring that the pollen record accurately reflects past regional vegetation changes

- Line 163 - main sum - I'm not quite sure what this means. The term "main sum" refers to the total pollen count used for percentage estimates, excluding Himalayan taxa, aquatic taxa, and spores, as described in the methods section.

- Line 163 - excluding Himalayan taxa – Why are there Himalayan taxa in the sample. Is the sediment really from the Mahanadi or perhaps the Ganges? Himalayan taxa are present in very low abundances in our samples likely due to minimal long-range transport (Clément et al., 2024, and references therein). However, multiple lines of evidence confirm that the primary source of the sediment at Site U1446 is the Mahanadi Basin rather than the Ganges-Brahmaputra system. The sedimentological characteristics, including elemental composition analyses from nearby IODP Site U1445, closely correspond to the lithology of the Mahanadi Basin. In addition, pollen transport in this region is dominated by fluvial input, with over 90% of the Mahanadi River's annual discharge occurring during the summer monsoon. Cross-shelf transport, favored by a narrow continental shelf and submarine canyons, ensures efficient delivery of sediment to the deep sea. In contrast, the Ganges-Brahmaputra plume is associated with significant flocculation at the river outlet, causing sediments to sink rapidly and limiting long-distance transport. Seasonal surface currents further reduce southward transport from the Ganges-Brahmaputra system. While a minor contribution from remote sources cannot be entirely ruled out, the extensive evidence supports the Mahanadi Basin as the dominant sediment source for Site U1446.

- Line 185 - primary indicator of ISM rainfall variability - But the monsoon is seasonal. Couldn't this be found in a tropical, non-seasonal environment? Line 238 - increasing dryness. – and seasonality? We agree with the reviewer that the monsoon is seasonal. However, in the core monsoon zone of India, 80%–90% of annual rainfall occurs during the summer monsoon (June–September) (Gadgil, 2003). In this sentence, we refer to the wet evergreen forest ecological group as the primary indicator of summer monsoon rainfall variability. This group includes wet evergreen, semi-evergreen and moist deciduous taxa, which thrive in the wettest forest formations of India due to their reliance on sustained high precipitation levels. Hence, their abundance in the sediments of Site U1446 is interpreted as reflecting summer monsoon rainfall variations.

- Line 196 - over the India's CMZ (17°-27.5° N, 67.5°-90° E) - over the Indian CMZ. Also mark this on the map figure. We will revise the sentence as suggested. The CMZ is already highlighted in Figure 1 with a striped area.

- Figure 2 – This chart is very hard to read. We will revise Figure 2 to enhance clarity by incorporating a color code for ecological groups and improving the graphical representation of low-abundance pollen taxa.

- Line 250 - The two forest expansion  - What does that mean? We acknowledge that the wording of 'the two forest expansion' could be clearer. We will rephrase it to specify that it refers to two distinct intervals of tropical forest expansion during MIS 11a, corresponding to Pollen Zones MIS 11a-1 and MIS 11a-3, as described in the Results section and highlighted in Figures 2 and 3.

- Line 300 - unusually high global terrestrial biosphere – Not sure what this means. Line 305 - contribution of tropical forests to the global terrestrial biosphere -  you did not estimate volumes of forest or the volumes of carbon that would be captured by the forest so I don't think you really understand what proportion of the global budget this represents. In line with our efforts to improve readability, we will remove this discussion to maintain focus on the key factors driving vegetation changes and monsoon dynamics during MIS 11. The term "unusually high global terrestrial biosphere" was originally used to describe the concept of "exceptionally high biosphere productivity" introduced by Brandon et al. (2020) to explain how an extensive terrestrial biosphere during MIS 11c may have delayed the $CO_2$ increase following the MIS 12/11 transition. However, since we did not quantify the global carbon or biomass contribution of forests and this aspect is beyond the scope of our study, we will remove it for clarity. This avoids unnecessary complexity and will make the manuscript more readable, as requested.

- Line 314 - asymmetric "M-shaped" structure – I'm unclear about what you mean. Are you talking about the shape of proxy record? Which one in particular? Yes, we are referring to the shape of the tropical forest percentage record during MIS 11, which exhibits two peaks, with the second being more pronounced, separated by a dip, forming an "M-shaped" pattern. We will clarify this more explicitly in the section. 4. Results and discussion.

- Line 325 - EDC ice core – what does EDC mean? And where was this core taken? EDC stands for EPICA Dome C, an ice core drilled in East Antarctica (Jouzel et al., 2007). We will clarify this in the revised manuscript.

- Line 327 - imprint on the CH4 record -  what record are you talking about? What is a proxy for methane? The $CH_4$ record refers to atmospheric methane concentrations reconstructed from ice cores. Methane ($CH_4$) levels are measured from trapped air bubbles in the ice, being used as a proxy for past variations in global methane emissions (e.g., Petit et al., 1999; Loulergue et al., 2004).

- Line 339 - tropical wetland extent – how can a speleothem record tell you about the extent of wetlands? We agree that a speleothem record is not a direct proxy for tropical wetland changes. In this sentence, we are referring to studies that have used the Sanbao speleothem record to infer wetland expansion. The underlying reasoning is that a northward shift of the Intertropical Convergence Zone led to a poleward intensification of Northern Hemisphere westerlies, creating favorable conditions for new tropical wetlands to form. This expansion resulted in increased methane production, which was reflected in methane records.

- Line 343 – speleothem data alone cannot be used as a definitive proxy - or that these data indicate complicated hydrological system. We entirely agree with the reviewer, and this point is extensively acknowledged in the manuscript, particularly in Section 4. Results and Discussion.

- Figure 5 - I suggest you use different colours when you're plotting two sets of data on the same plot. Using two shades of blue is a little confusing. Why don't you just use something more distinctive? As suggested, we will adjust the color scheme in Figure 5 to ensure clearer distinction between the datasets.

- Line 381 - The asymmetric M-shaped pattern – I don't know what you mean. You're going to need to label that quite clearly on one of your figures. This overlaps with the earlier remark on the 'M-shaped' pattern. As noted in our response to Line 314, we will clarify this in the text considering the reviewer's feedback. However, in line with our efforts to simplify Figures 4 and 5, we will not add additional visual elements to these figures.

- Line 386 – NHSI - This is a confusing abbreviation. If you told us what it was earlier in the manuscript, then I've already forgotten. I suggest you don't use this. "NHSI" stands for Northern Hemisphere Summer Insolation. As recommended, we will reduce unnecessary abbreviations throughout the manuscript; however, we will retain NHSI as it is a commonly used term in paleoclimate studies and provides a concise way to refer to this key forcing factor.

- Line 426 - the absence of ice-sheet forcing and associated freshwater flux in the simulations - this seems like a major omission to me. Is it necessary to ignore the ice sheets in the model? Seems like a bad model. We understand the reviewer's concern regarding the absence of ice-sheet forcing and associated freshwater fluxes in the model simulations. As noted in our previous response regarding the model caveats, our study focuses on isolating the impact of insolation and $CO_2$ during the MIS-11 optimum. However, we recognize that excluding ice sheets and freshwater flux may contribute to the mismatch between model outputs and proxy records during the glacial inception. This discrepancy further underscores the potential role of these factors in triggering abrupt climate events, reinforcing the need for future simulations that incorporate interactive ice-sheet dynamics, as stated in the manuscript.

- Line 444 - Yongxing Cave -but that is east Asian monsoon and not entirely clear that it's appropriate. The reviewer is correct that Yongxing Cave, located in China, is influenced by the East Asian monsoon, as shown in Figure 1. In this sentence, we refer to a regional-scale comparison rather than changes strictly within the Core Monsoon Zone of India. The high-resolution speleothem record from Yongxing Cave provides a broader context for assessing monsoon variability and its potential link to the abrupt savannization event in India.

- Line 472 - involves distinct in-cave processes – also, they may be issues related to regional rainfall pattern. "We thank the reviewer for pointing this out. However, we did not address this aspect because the East Asian speleothem records discussed in our study are geographically very close to each other, as shown in Figure 1, and are, in principle, influenced by the same regional rainfall patterns.

- Line 541 - the asymmetric M-shaped pattern in our tropical forest record -I never really understood what you meant by this. This overlaps with a previous comment on the 'M-shaped' pattern. As noted in our response to Line 314, we will clarify this further in the text.

- Line 555 - $CH_4$ overshoots – I don't understand what that means. "Overshoots" is commonly used in paleoclimate research to describe sharp rises in proxy records. In our study, it refers to high-amplitude methane increases as identified by Nehrbass-Ahles et al. (2020).

Minor corrections such as typos, abbreviations, references and small changes will be carried out to improve the readability of the text.

Line 73 – IODP – Define abbreviation. And SST.

Line 79 - during MIS 5e - provide age in ka.

Line 101 - Integrated Ocean Drilling Program (IODP)- Say this on first use.

Line 106 - LOVECLIM Earth System Model – Need a reference here.

Line 111 - δ18Ob – Superscript "18". Is "b" right? "

Figure 1 -  Text in inset North Atlantic map needs to have the same font as the rest of the figure. Text needs to be larger.

Line 154 - A total of 62 levels were sampled for pollen analysis, - A total of 62 samples were taken for pollen analysis.

Line 171 - tropical forest (TF)  - Not sure this abbreviation is necessary. Line 174 - tropical forest (TF) – If you are going to abbreviate you only need to define this once.

.Line 189 - Fig. 4 -  Call out figures in numerical order.

Line 231 - abundance tropical deciduous forest – abundance of tropical deciduous forest.

Line 246 - PZ MIS 11a-1,  - You use too many abbreviations. I already forgot what PZ means.

Line 250 - substantial less humid condition - substantially less humid conditions.

Line 258 - decrease of 22.3% - decrease to 22.3%?.

Line 296 - long records – long duration records.

Line 395 - gradually declines - gradually declined.

Line 483 – soutprashern -  Spelling.

Line 498 - still highly discussed  - strongly debated? Controversial?

Line 505 - carbon dioxide jumps (CDJ) - this is entirely inappropriate. You have far too many abbreviations. The manuscript is extremely hard to read anyway without you adding this. Nobody understands these abbreviations when there are so many.

Line 503 - DO-like  - More confusion.

Line 526 - Ice-Rafted Detritus (IRD)  - Please stop.

Line 548 - IN the CH$_4$ record -  IN?

**References:**

– Brandon, M., Landais, A., Duchamp-Alphonse, S., Favre, V., Schmitz, L., Abrial, H., Prie, F., Extier, T., & Blunier, T. (2020). Exceptionally high biosphere productivity at the beginning of Marine Isotopic Stage 11. Nature Communications, 11, 2112. DOI: 10.1038/s41467-020-15739-2.

– Clément, C., Martinez, P., Yin, Q., Clemens, S., Thirumalai, K., Prasad, S., Anupama, K., Su, Q., Lyu, A., Grémare, A., & Desprat, S. (2024). Greening of India and revival of the South Asian summer monsoon in a warmer world. Communications Earth & Environment, 5, 685. DOI: 10.1038/s43247-024-01781-1.

– Gadgil, S. (2003). The Indian monsoon and its variability. Annual Review of Earth and Planetary Sciences, 31, 429–467. DOI: 10.1146/annurev.earth.31.100901.141251.

– Hooghiemstra, H., Lézine, A.-M., Leroy, S. A. G., Dupont, L., & Marret, F. (2006). Late Quaternary palynology in marine sediments: A synthesis of the understanding of pollen distribution patterns in the NW African setting. Quaternary International, 148, 29–44. DOI: 10.1016/j.quaint.2005.11.007.

– Jouzel, J., Masson-Delmotte, V., Cattani, O., Dreyfus, G., Falourd, S., Hoffmann, G., Minster, B., Nouet, J., Barnola, J. M., Chappellaz, J., Fischer, H., Gallet, J. C., Johnsen, S. J., Leuenberger, M., Loulergue, L., Lüthi, D., Oerter, H., Parrenin, F., Raisbeck, G., Raynaud, D., Röthlisberger, R., Schmidt, A., Schilt, A., Schwander, J., Selmo, E., Souchez, R., Spahni, R., Stauffer, B., Steffensen, J. P., Stenni, B., Stocker, T. F., Tison, J. L., Werner, M., & Wolff, E. W. (2007). Orbital and millennial Antarctic climate variability over the past 800,000 years. Science, 317, 793–796. DOI: 10.1126/science.1141038.

– Loulergue, L., Parrenin, F., Blunier, T., Barnola, J. M., Spahni, R., Schilt, A., Raisbeck, G., Jouzel, J., & Chappellaz, J. (2008). Orbital and millennial-scale features of atmospheric $CH_4$ over the past 800,000 years. Nature, 453, 383–386. DOI: 10.1038/nature06950.

– Nehrbass-Ahles, C., Shin, J., Schmitt, J., Bereiter, B., Joos, F., Schilt, A., Schmidely, L., Silva, L., Teste, G., Grilli, R., Chappellaz, J., Hodell, D., Fischer, H., & Stocker, T. F. (2020). Abrupt $CO_2$ release to the atmosphere under glacial and early interglacial climate conditions. Science, 369, 1000–1005. DOI: 10.1126/science.aay8178.

– Petit, J. R., Jouzel, J., Raynaud, D., Barkov, N. I., Barnola, J. M., Basile, I., Bender, M., Chappellaz, J., Davis, M., Delaygue, G., Delmotte, M., Kotlyakov, V. M., Legrand, M., Lipenkov, V. Y., Lorius, C., Pepin, L., Ritz, C., Saltzman, E., & Stievenard, M. (1999). Climate and atmospheric history of the past 420,000 years from the Vostok ice core, Antarctica. Nature, 399, 429–436. DOI: 10.1038/20859.

– Raman, C. V., and Reddy, K. S. N. (2001). Sediment dispersal pattern off the Mahanadi-Nagavali continental shelf, Northwest Bay of Bengal. Journal of the Geological Society of India, 58, 123–133.

– Zorzi, C., Desprat, S., Clément, C., Thirumalai, K., Oliveira, D., Anupama, K., Prasad, S., & Martinez, P. (2022). When Eastern India oscillated between desert versus savannah-dominated vegetation. Geophysical Research Letters, 49, e2022GL099417. DOI: 10.1029/2022GL099417-

---

## Author Comment (AC2)

**Authors' Response to Reviewer #2:**

Oliviera and co-authors present interesting new data from an important interglacial period that may be analogous to the Holocene. I believe the data is of good quality and the topic is interesting. But I found the writing and figures of the paper to be overly complicated and difficult to follow. I would suggest a major simplification and focusing on key takeaways in both the text and figures so that readers with a wide range of interests can learn something from the paper. As it is currently written, one needs to be an expert in the Indian summer monsoon and MIS 11 to be able to understand the presentation. But in order to reach the goal of demonstrating how this era could be relevant to the Holocene and future warming, the presentation needs to be simplified. I will provide some examples and ideas of how to clarify, mainly focused on figures and nearby text, but please work on the paper as a whole to simplify the message and improve readability.

Overall, I would suggest a focused rewrite and figure updates that highlight key takeaways, but at the same time acknowledges that it is complicated. It's a big win to bring together all of these different data sets, complexity is more exciting to me than a simple answer.

We appreciate the reviewer's constructive comments and positive assessment of our study. We acknowledge that certain parts of the manuscript may be overly detailed, which could make it more challenging for a broader audience to follow. As suggested, we will undertake a focused revision to enhance readability and highlight key takeaways. We will simplify and shorten the text, particularly in Section 4: Results and Discussion, and reduce abbreviations throughout the manuscript while ensuring that the interplay of forcings driving Indian vegetation and summer monsoon variability during MIS 11 is clearly presented. As the reviewer highlights, integrating multiple datasets to analyze these interactions is a key strength of this study. We will ensure that our findings are effectively presented to the broader readership of Climate of the Past.

We will also revise the figures to enhance clarity and readability. Below, we address each suggested change.

Relating to figure 1: The maps as currently presented are confusing. I would recommend breaking this into two separate panels rather than having the Atlantic sites inset. Next, I would add another panel with a figure or table summarizing the preceding text which has a lot of information about rainfall amounts and vegetation type that would be better presented graphically. As suggested, we will revise Figure 1 to enhance clarity, including adjustments to the layout, improvements to the vegetation information and the separation of the Atlantic and Indian regions into two panels.

Relating to figure 2: I know that it is valuable to show the whole pollen diagram, but I would love some additional context and explanation for which taxa are important for the green bands. I see some patterns but its tough to get a cohesive understanding. We acknowledge the need for a clearer representation of the pollen diagram; hence, as recommended, we will revise it for improved readability and interpretation by incorporating a color code and enhancing the graphical visualization of very low-abundance pollen taxa.

Figure 3. The combination of the arrows, the dashed lines, and the shaded bars is a lot to take in. I have a tough time knowing what I am supposed to be looking at in this figure. We will simplify the visual elements to enhance readability and ensure that key patterns are clearly highlighted.

Figure 4. It's a bit tough for me to understand why the highlighted sections are important, but other similar large magnitude changes in the pollen are not interpreted. We understand the reviewer's concern and will clarify the criteria used to identify forest expansion and contraction events, represented by the green and brown bands, respectively, in the figure caption.

Figure 5. The data should be in front of the bars on the graph. As it is currently, the bars obscure the data. To my eye, it's not totally clear that NHSI is the main driver of the dynamics as it seems like the text is saying. The patterns look like they share features of lots of the potential drivers plotted in this figure. We will adjust the figure layout to ensure that the data is displayed in front of the bars for better visibility and to further clarify the contributions of the main forcings to the observed variability.

Finally, we note that Table 1 provides a detailed interpretation of the pollen record, allowing for a more concise presentation of the results in the figures and text.